# Research on Sustainable Development Evaluation of Reservoir Water Source Area in Island Area

**Jo-Hung Yu** [1]**, Hsiao-Hsien Lin** [2,3,*] **, Yu-Chih Lo** [2]**, Kuan-Chieh Tseng** [4] **and Chin-Hsien Hsu** [2,*]

[1] Department of Marine Leisure Management, National Kaohsiung University of Science and Technology, Kaohsiung 811213, Taiwan; henry@nkust.edu.tw

[2] Department of Leisure Industry Management, National Chin-Yi University of Technology, Taichung 41170, Taiwan; loyuchih@ncut.edu.tw

[3] Tourism Managemen, Athena lnstitute of Holidtic Wellness, Wuyi University, No 26, Wuyi Avenue, Wuyishan 354300, China

[4] MA Program in Social Enterprise and Cultural Innovation Studies, College of Humanities & Social Sciences, Providence University, Taichung 43301, Taiwan; jackt72@pu.edu.tw

* Correspondence: chrishome12001@yahoo.com.tw (H.-H.L.); hsu6292000@yahoo.com.tw (C.-H.H.)

**Abstract:** This study takes Taiwan's Sun Moon Lake Reservoir as the research object to discuss the sustainable development decision-making of the reservoir water source area in the island area. The grounded theory was adopted to construct the framework; 835 questionnaires were analyzed by statistical test; 10 interviewees' feelings were collected through interviews and discussions, and multiple verification methods were used for exploration. Conclusion: The study found that the current development of the reservoir water source area of the outlying islands has brought much garbage; affecting the quality of the natural environment; lake; and water source; caused the disappearance of distinctive culture and architecture. The poor interaction between businesses and communities has led to an overlap in the types of industries, consumer goods, attractions and a lack of transportation and medical facilities, affecting people's desire to travel. The government's future decisions include: Increasing the variety of consumer goods; improving medical and transportation facilities; preserving unique culture and architecture; linking foreign-invested enterprises with community interactivity; enhancing local people's interaction; and compensating for the deficiencies in human resources for industrial development are the key points for future improvement of the reservoir water source area of the outlying islands.

**Keywords:** outlying island reservoir; experiential value; tourism choice factor; tourism behaviors; Sun-Moon Lake



## 1. Introduction

Water and soil are the main resources on which human life depends. With the demands of human development, water and soil resources are exploited and utilized. Tourism is based on the rich natural landscape, diverse ecological environment, long history and culture, magnificent monuments and buildings, different economic crops and other sightseeing resources, combined with residents' human and material inputs, to form tourist reception services and attract tourists to travel and consume. All countries look forward to the interaction of tourism activities and industrial development to promote developing local economy and industries. Tourism is a kind of industry, phenomenon and behavior [1].

Water and soil resources, when applied to developing the tourism industry, require proper resource management. Good water and soil resource management can preserve local natural ecology and achieve the goal of resource sustainability. This represents the management performance of the local people and proves the results of sustainable development of local tourism. With good management measures for sustainable development, resources can be sustainable, and tourism attraction can be enriched, which is one of the factors influencing tourists' motivation to travel [2] and their intention to travel. Therefore,

it is important to understand the experience of visitors and evaluate the effectiveness of local development. However, feelings usually run high after the experience of tourism activities [3–5]. Understanding consumer psychology can help tourism areas or industries deliver product information [6]. It can so explore the effectiveness of local development and the tourists' attitudes and feelings. It has become a measurement method for market segmentation [7].

Attitude is a behavioral image often explored in the field of psychology, social science and marketing. It is an important explanatory variable for predicting mentality, developed from marketing and sociology [8], and has been extended to predict the relationship between consumers and products. Tourists are also consumers and tourism experience in consumer behavior [9]. Tourism can be considered as an activity, as well as consumer behavior and a phenomenon. Therefore, understanding tourists' consumer behavior can be studied from the perspective of consumer behavior. Tourists' consumer behavior is a derivative of consumer attitudes. After the inner demand is generated, tourist activities and consumption phenomena produce the consumption pattern of tourists through actual consumption experience and cognition [10]. It can be used to understand whether the effectiveness of local tourism development meets the needs of tourists [11].

The reason why the tourism phenomenon occurs is due to the tourism motivation of the people, coupled with the attraction of tourism resources, causing tourists to stay away from their homes, leave their home ranges, and engage in activities unrelated to their work, to satisfy personal physical or psychological needs, improve their physical conditions, and broaden their minds. However, although people expect that tourism can satisfy them [10–12], the current economic, social, and environmental conditions generated by different residences may affect an individual's perception of quality of life [13], and different perceptions of quality of life will affect personal attitudes towards leisure and behavior [14]. The differences in personal attitudes and behaviors will produce different individual needs and motivations [15]. Different needs and motivations make people have different views on individual experience and product acceptance [16]. Understanding tourists' feelings after their consumption experience are helpful for the locals to correct defects [17].

Taiwan is an island. Although Sun Moon Lake is located in the central mountainous region of Taiwan, with various stages of development and management, it has perfected the functions of leisure sports, living and dining, irrigation and farming, and technological power generation, and is an important water resource area in Taiwan. It is also a tourist attraction that makes full use of the local water and the natural resources of the surrounding mountains and has become a multifunctional reservoir that is currently the top choice for domestic tourism, attracting many tourists to the area every year. In 2018, more than 830,000 tourists visited Sun Moon Lake, with an average daily consumption of approximately USD 221.76 per person, bringing in business revenues of USD 184 million [14]. Because of geographical factors, Taiwan is divided into northern, central, southern, eastern regions and outlying islands. The north is technologically advanced and fast-paced; the central part is rich in agricultural products and diverse cuisines; the south is industrially developed and rich in marine resources; the east is ecologically diverse, slow, and leisurely; and the outlying islands are surrounded by the sea, with exquisite land and simple folklore. Although, in general, there are abundant natural ecological and tourism resources, coupled with developing technology and the improvement of living standards, the public's awareness of tourism has increased, and their attitude towards tourism is positive. However, compared with the historical peak, the number of tourists has not risen but has been falling year-by-year, with a decline of 768.6% [18], which shows that a bottleneck in developing the region has emerged, and the people of each region may have a common or different feeling about the effectiveness of developing tourism in the region, so developing the region is no longer fully favored by the public. This is undoubtedly a blow to island-type regions that lack material and mineral resources and expect to improve local growth through developing multifunctional economic behaviors using reservoir water resources.

The development of water source areas is a model for expanding the tourism industry by combining local water and soil resources with other features and resources. It is more popular with the public and is currently the mainstream of tourism activity planning. However, tourism development is all-round, and the impact level can be viewed from the economic, social, and environmental levels [18,19]. The economy will be explored from the perspectives of cost of living, industrial construction and village development [2], and then the employment, salary, consumption, construction, industry, facilities, prices, discounts, health, cultural and creative activities, recreational activities, community feedback and decision-making coordination issues [2–6]. Viewing the society from the perspectives of tourism facilities, community construction, living atmosphere, culture, and public security [2] is necessary for us to understand the popularity, service and activity quality, policy participation, tourism organization planning, cultural and architectural features, public security maintenance, community construction, and people Interaction and other issues [14,15,20]. Viewing the environment from the perspectives of tourism and recreational facilities, natural ecology, etc. [2] is necessary for us to explore public transportation, parking and recreational space, environmental literacy of tourists, garbage volume, woodland and ecological habitat, fumes from motorcycles and cars, water sources and air quality [2,21–23]. Therefore, exploring people's travel experiences and analyzing their travel attitudes from the perspective of their different places of residence can estimate their travel preferences and behavior patterns to facilitate adjustment of development decisions [2,24] and overcome local development difficulties in response to tourist needs.

Usually, tourism development research is based on the characteristics of tourists [25,26], different places of residence [27,28], and tourism experience [29,30] or using spatial-visual angle [31,32] to explore the effectiveness of tourism development. However, the current development of island tourism has gradually begun to show defects, tourism attractiveness has declined, the tourism environment has gradually deteriorated, and the tourism space has been gradually reduced [33,34]. Moreover, under different spatial backgrounds, public facilities planning, economic development scale and consumption levels will have different degrees of development. Effectiveness difference [35]. It will affect the basic quality of life and needs of the individual, the motivation and behavior of the tourism industry, and lead to different levels of tourism needs and feelings [36]. Therefore, the researchers believe that, based on the personal characteristics formed by different geographic information, from a spatial perspective, explore the development status of the water source of the outlying island reservoir and find out the problems that need improvement. It is conducive to maintaining the existing environmental conditions and developing in the direction of sustainable development.

## 2. Methods and Instruments

The researcher believes that although there are many studies on tourism behaviors and tourism development experience [3–25], few of them have been carried out to survey people's tourism behaviors and attitudes in certain tourist area based on their experience in the tourist area from the perspective of personal characteristics formed by different geographical information and from the space angle. If the research participants or field has not been discovered or explored, the theory is constructed based on the collected data [32]. By means of grounded theory, the problem to be solved will be grasped based on the same data obtained from the same situation [37]. Therefore, based on the existing research theoretical basis [3–25], the researcher continues the argumentation and inference results [32] and uses the experience of different life backgrounds to examine the current development of the water source area of the island reservoir. It is expected that by integrating the results of various studies and suggestions, we will be able to make sustainable decisions on water and environmental conservation in areas where water is scarce and multifaceted development of water resources is desired.

Therefore, the grounded theory was adopted to construct the research framework of this study. From July to September 2019, the field survey method was used for observation,

and convenience sampling was adopted to distribute 835 questionnaires. Next, SPSS for Window 22.0 software, ANOVA test method, and descriptive verification method were used for analysis. Then, 10 residents, tourists and scholars and experts, coded as AJ, were interviewed. The messages and information conveyed by the interviewees were recorded in the way of recording notes. After this, all the data were inducted, organized, and analyzed to construct a research paper [22]. Finally, a multiple verification analysis method was adopted, combined with the information of different research participants, research theories and methods, from multiple points of view, to verify multiple data and compare all the results [38–40] to obtain accurate knowledge and meaning. As shown in Figure 1.

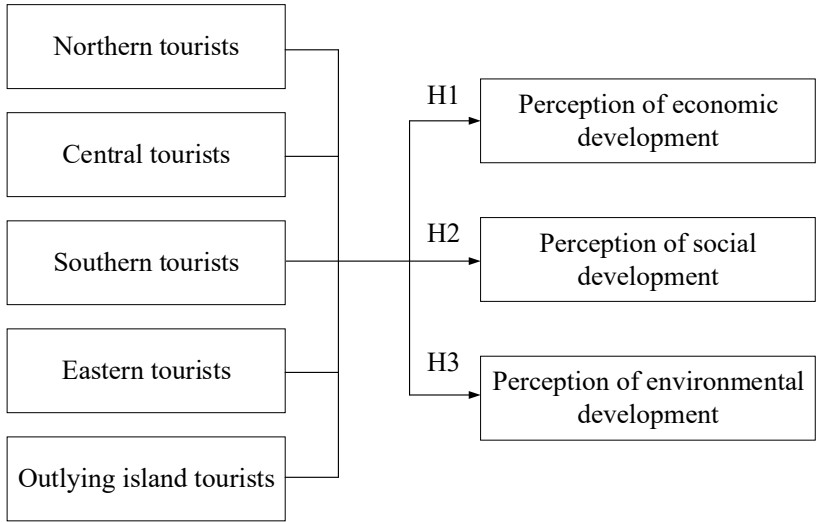

**Figure 1.** Study framework.

After consulting references and related research results [3–25], compiling and editing questionnaire tools, analyzing the economic, social, and environmental dimensions, it is found that the Cronbach'α value is 0.90–0.97, and the value is greater than 0.9, indicating that the questionnaire is highly reliable [41]. As shown in Table 1.

**Table 1.** Analysis of factors in the perception of impacts on tourism.

| Facet | Second Facet | Issues | Cronbach'α |
|---|---|---|---|
| Economic | Civil price | Employment opportunities, entrepreneurial opportunities, salary income, expenditure costs | 0.90–0.91 |
| | Industry construction | Tourism construction, tourism industry, the combination of characteristics, leisure opportunities, tourism discounts | |
| | Village development | Facilities maintenance, development feedback, land and house prices, healthcare, community communication, protection policies, policy participation, cultural and creative products, interpretation facilities | |
| Society | Community building | Popularity, service quality, event quality, community participation, sufficient indicators, recreational options, increased organization | 0.96–0.97 |
| | Atmosphere of life | Youth return home, vocational training opportunities, cultural preservation, architectural features, tourist attitudes, living environment, corporate image, cultural industry, cultural activities, community beautification, quality of life | |
| | Cultural security | Public interaction, public security management, repurchasing property, corporate feedback, community autonomy | |
| Environment | Recreational facilities | Living space, garbage resettlement, tourism transportation, parking and rest, tourist destruction, development area increase | 0.91–0.92 |
| | Tourist facility | Route planning, tourism transportation, transportation connections, bicycle paths, trails, Wi-Fi facilities, bicycle rental | |
| | Ecosystem | Environmental literacy, nature conservation, historic site preservation, community cleaning, littering | |
| | Conservation measures | Water quality, appearance of vegetation, appearance of habitat, oily fume and exhaust gas, noise garbage, external ecological threats | |
| | Natural landscape | Air quality, lake water quality | |

## 3. Research Results and Analysis

### 3.1. Background Information Analysis

According to the analysis, most tourists are female, mainly living in the central region of Taiwan, and most of them are 21–30 years old. They plan to travel once a year on average, taking a one-day self-guided tour for recreational sports. As shown in Table 2.

**Table 2.** Analysis of tourist backgrounds.

| Name | Issue | % | Name | Issue | % |
|---|---|---|---|---|---|
| Address | North<br>Central<br>South | 7.6%<br>84.2%<br>5.6% | Address | East<br>Island | 1.9%<br>0.7% |
| Gender | Male<br>Female | 45.6%<br>54.4% | Age | Under 20<br>21–30 | 27.6%<br>40.9% |
| Age | 51–60<br>Over 61 | 4.4%<br>1% | | 31–40<br>41–50 | 14.6%<br>11.9% |
| Travel days | 1 day<br>2–3 day<br>4–5 day<br>6–7 day | 68.8%<br>20.9%<br>3.7%<br>2.6% | Travel mode | Free travel<br>Family travel<br>Group travel<br>Teaching activities | 45.2%<br>39.1%<br>16.7%<br>2.9% |
| Travel purpose | Academic survey<br>Education training<br>Official activities<br>Jobs<br>Shopping<br>Leisure and sports<br>Tourism | 3.1%<br>18%<br>2.8%<br>2.8%<br>18.7%<br>20%<br>54.3% | Travel frequency | Official visit<br>Once a week<br>Once a month<br>Once every six months<br>Once a year<br>More than 1 year | 2.6%<br>3.5%<br>6%<br>6%<br>24.2%<br>52.6% |

### 3.2. Cognitive Analysis of Tourists' Rural Experiences

The analysis of tourists' current attitudes to travel and influencing factors [15–17] can be conducted based on their travel experiences [28,29], so first use statistical verification to analyze the people's experience of the development status of the water source area of outlying island reservoirs, and To find out the effectiveness and difficulties of development.

On the whole, cultural and creative products, tourism industry, land price, cultural preservation, community Involvement, public interaction, bicycle rental, environmental literacy, lake water quality, etc., currently maintain good results. However, there is still insufficient community communication, tourism construction, salary income, youth return home, service quality, public security management, Wi-Fi facilities, threats of alien species, community clean, park and rest, air quality, noise garbage, tourist vandalism, lake water quality. The development of such issues is not effective and needs to be resolved. As shown in Table 3.

**Table 3.** Cognitive analysis of tourist rural experiences.

| Facet | Second Facet | High Perception | Low Perception |
|---|---|---|---|
| Economic | Civil price<br>Industry construction<br>Village development | Cultural and creative products<br>Tourism industry<br>Land price | Community communication<br>Tourism construction<br>Salary income |
| Society | Community building<br>Atmosphere of life<br>Cultural security | Cultural preservation<br>Community Involvement<br>Public interaction | Youth return home<br>Service quality<br>Public security management |
| Environment | Recreational facilities<br>Tourist facility<br>Ecosystem<br>Conservation measures<br>Natural landscape | Bicycle rental<br>Noise garbage<br>Environmental literacy<br>Tourist vandalism<br>Lake water quality | Wi-Fi facilities<br>Alien ecological threat<br>Community clean<br>Park and rest<br>Air quality |

Research inference, because the locals have protected culture, developed cultural and creative products, and attracted a large amount of private investment, tourism has been developed rapidly, stimulating local economy and development. However, the existing tourist facilities and parking spaces are insufficient, the population is gradually aging, and industries are short of manpower [5]. These problems have hindered developing tourism, and there is a gap in scenic area management.

### 3.3. Analysis of Cognitive Differences in Rural Experiences of Tourists from Different Regions

Approaching the issue from the perspectives of different residential areas [26–29], we can study tourists' travel behavior from different regions and obtain the perception of local development in different regions. Therefore, in this study, the ANOVA method was used to analyze the rural experience cognition of tourists from different regions to verify whether there are differences in the scenic area cognition of tourists from different regions [26–31].

#### 3.3.1. Cognition of Current Economic Development

The analysis found that people in the north have high feelings about entrepreneurial opportunities (3.71), feature combination (3.81), cultural and creative products (4.0). The central part has a high sense of entrepreneurial opportunities (3.82), feature combination (3.9), cultural and creative products (3.86). The south has a high feeling of entrepreneurial opportunities (3.62), tourism industry (3.91), cultural and creative products (3.89). The eastern part of the country has high feelings about job opportunities (3.94), tourism industry (4.38), healthcare and interpretation facilities (3.75). Outlying islands to job opportunity and entrepreneurial opportunities (4.0), tourism industry and sightseeing discount (4.33), healthcare and community communication (4.5) high. As shown in Table 4.

**Table 4.** Analysis of economic development cognition of tourists from different regions.

| Issue | M | | | | | F | p | Scheffé/Kruskal–Wallis |
|---|---|---|---|---|---|---|---|---|
| | North | Central | South | East | Island | | | |
| Job opportunity | 3.32 | 3.75 | 3.47 | 3.94 | 4 | 4.52 | 0.00 * | NS |
| Entrepreneurial opportunities | 3.71 | 3.82 | 3.62 | 3.38 | 4 | 1.78 | 0.13 | NS |
| Salary income | 3.4 | 3.47 | 3.43 | 3.38 | 3.5 | 0.14 | 0.97 | NS |
| Expenditure cost | 3.6 | 3.56 | 3.49 | 3 | 3.67 | 1.46 | 0.21 | NS |
| Tourism construction | 3.73 | 3.78 | 3.36 | 3.75 | 4.17 | 2.45 | 0.05 | NS |
| Tourism industry | 3.71 | 3.85 | 3.91 | 4.38 | 4.33 | 2.24 | 0.06 | NS |
| Feature combination | 3.81 | 3.9 | 3.77 | 3.94 | 4.17 | 0.48 | 0.75 | NS |
| Leisure opportunity | 3.57 | 3.88 | 3.83 | 3.81 | 4.17 | 2.07 | 0.08 | NS |
| Sightseeing discount | 3.62 | 3.81 | 3.74 | 3.5 | 4.33 | 1.72 | 0.14 | NS |
| Facility maintenance | 3.78 | 3.79 | 3.57 | 3.75 | 4.17 | 0.89 | 0.47 | NS |
| Development feedback | 3.65 | 3.78 | 3.83 | 3.25 | 3.83 | 1.81 | 0.12 | NS |
| Healthcare | 3.43 | 3.63 | 3.74 | 3.75 | 4.5 | 2.41 | 0.05 | NS |
| Interpretation facilities | 3.65 | 3.8 | 3.66 | 3.75 | 4.17 | 0.86 | 0.49 | NS |
| Cultural and creative products | 4 | 3.86 | 3.89 | 3.06 | 3.67 | 3.41 | 0.00 * | NS |
| Community communication | 3.59 | 3.64 | 3.55 | 3 | 4.5 | 3.42 | 0.00 * | Island > East |
| Protection policy | 3.63 | 3.75 | 3.55 | 3.69 | 4.17 | 0.96 | 0.43 | NS |
| Policy participation | 3.6 | 3.69 | 3.64 | 3.31 | 3.67 | 0.79 | 0.53 | NS |

\* $p < 0.01$.

The study concluded that due to differences in the economic structure and development status of tourists' different places of residence, coupled with different sensitivity to policy participation, differences in economic development, sanitary conditions and living standards, there was a difference ($p < 0.01$) in the perceptions of tourists from different regions about employment opportunities, cultural and creative products, and community communications, especially those from offshore islands.

### 3.3.2. Cognition of Current Social Development

The analysis found that people in the north have high feelings about community involvement (3.03), cultural activities (3.24), public interaction (3.19). The central part has a high sense of public Security management (2.89), cultural preservation (2.9), cultural and creative products (3.86). The south has a high feeling of public interaction (3.51), building style and cultural activities and vocational training opportunities (3.3), cultural and creative products (3.89). The eastern part of the country has high feelings about community involvement (3.44), cultural preservation (3.56), public interaction (3.31). Outlying islands to reputation and sufficient indicators (3.5), cultural preservation (3.67), public Security management (4.0) high. As shown in Table 5.

**Table 5.** Analysis of cognition of social development of tourists from different regions.

| Issue | M | | | | | F | *p* | Scheffé/Kruskal–Wallis |
|---|---|---|---|---|---|---|---|---|
| | North | Central | South | East | Island | | | |
| Reputation | 2.89 | 2.66 | 3.49 | 3.31 | 3.5 | 0.55 | 0.00 * | South > Central |
| Activity quality | 2.9 | 2.68 | 3.34 | 3.06 | 2.83 | 0.06 | 0.00 * | South > Central |
| Service quality | 2.75 | 2.74 | 3.34 | 3.06 | 3.17 | 0.28 | 0.00 * | South > Central |
| Community involvement | 3.03 | 2.82 | 3.38 | 3.44 | 2.83 | 0.65 | 0.01 * | NS |
| Sufficient indicators | 2.9 | 2.89 | 3.36 | 3.19 | 3.5 | 0.15 | 0.05 | NS |
| Recreation options | 2.97 | 2.82 | 3.11 | 3.31 | 3.33 | 0.88 | 0.12 | NS |
| Organization increase | 2.62 | 2.87 | 3.17 | 3.5 | 3 | 0.17 | 0.01 | NS |
| Building style | 3.16 | 2.73 | 3.3 | 3.38 | 3 | 0.4 | 0.00 * | South > Central |
| Living environment | 2.92 | 2.82 | 3.15 | 3.25 | 3.33 | 0.1 | 0.12 | NS |
| Corporate image | 2.98 | 2.78 | 3.06 | 2.75 | 3 | 0.11 | 0.3 | NS |
| Culture industry | 3.1 | 2.87 | 3.23 | 2.63 | 3.17 | 0.55 | 0.12 | NS |
| Cultural activities | 3.24 | 2.83 | 3.3 | 3.19 | 3.33 | 0.69 | 0.00 * | South > Central |
| Cultural preservation | 2.83 | 2.9 | 3.19 | 3.56 | 3.67 | 0.36 | 0.03 | NS |
| Vocational training opportunities | 2.97 | 2.89 | 3.3 | 3.06 | 3 | 0.14 | 0.17 | NS |
| Quality of life | 2.79 | 2.87 | 3.15 | 3.31 | 2.67 | 0.55 | 0.00 * | NS |
| Community beautification | 3.13 | 2.85 | 3.19 | 3.19 | 3.17 | 0.28 | 0.00 * | NS |
| Repurchase | 2.94 | 2.86 | 3.21 | 2.75 | 3 | 0.33 | 0.36 | NS |
| Public security management | 3.11 | 2.89 | 3.43 | 3.19 | 4 | 0.54 | 0.00 * | NS |
| Public interaction | 3.19 | 2.77 | 3.51 | 3.31 | 3.83 | 0.43 | 0.00 * | NS |
| Community autonomy | 2.98 | 2.87 | 3.34 | 3.13 | 3.17 | 0.84 | 0.09 | NS |
| Corporate feedback | 3.14 | 2.86 | 3.36 | 3 | 3 | 0.06 | 0.00 * | NS |

\* $p < 0.01$.

The difference in social and cultural structures and the uneven degrees of international cultural exchanges and contacts have created differences in the cognition of social development experiences among tourists living in different places of residence. In addition, there were differences in personal cultural perceptions, requirements of quality of life, and different degrees of involvement in the scenic area, so there was a difference ($p < 0.01$) in the perceptions of tourists living in different places of residence about popularity, activities and service quality, architectural features, and cultural activities, especially those from the southern region had deepest perceptions.

### 3.3.3. Cognition of Current Environmental Development

The analysis found that people in the north have high feelings about garbage resettlement (3.79), Trail (3.97), littering (3.73), foreign threat (3.75), and air quality (3.71). The central part has a high sense of living space (3.64), traffic connection (3.66), littering (3.66), water source (3.72), air quality (3.72). The South has a high feeling of living space (3.69), bicycle rental and tourist transport (3.88), environmental literacy (3.44), fume exhaust (3.69), air quality and lake water quality (3.0). The eastern part of the country has high feelings about living space (3.69), bicycle rental and tourist transport (3.83), environmental literacy (3.56), fume exhaust (3.69), air quality and lake water quality (3.0). Outlying islands

to garbage resettlement (4.0), traffic connection, bicycle lane, Wi-Fi facilities and tourist transport (4.0), environmental literacy and littering (3.83), noise garbage (4.33), air quality and lake water quality (3.67) high. As shown in Table 6.

**Table 6.** Analysis of cognition of tourists from different regions about the current status of environmental development.

| Issue | M | | | | | F | p | Scheffé/Kruskal–Wallis |
|---|---|---|---|---|---|---|---|---|
| | North | Central | South | East | Island | | | |
| Living space | 3.6 | 3.49 | 3.64 | 3.69 | 3.33 | 0.44 | 0.78 | NS |
| Tourist vandalism | 3.35 | 3.36 | 3.45 | 3.5 | 3.67 | 0.22 | 0.93 | NS |
| Increased development area | 3.56 | 3.56 | 3.53 | 3.56 | 3.33 | 0.08 | 0.99 | NS |
| Parking and rest | 3.54 | 3.72 | 3.36 | 3.56 | 3.67 | 1.98 | 0.1 | NS |
| Garbage resettlement | 3.79 | 3.65 | 3.55 | 3.63 | 4 | 0.64 | 0.64 | NS |
| Trail | 3.97 | 3.79 | 3.62 | 3.31 | 3.83 | 1.91 | 0.11 | NS |
| Route planning | 3.92 | 3.64 | 3.53 | 3.63 | 3.83 | 1.47 | 0.21 | NS |
| Traffic connection | 3.68 | 3.43 | 3.66 | 3.63 | 4 | 1.63 | 0.16 | NS |
| Bicycle lane | 3.89 | 3.66 | 3.53 | 3.69 | 4 | 1.31 | 0.26 | NS |
| Bicycle rental | 3.76 | 3.64 | 3.45 | 3.88 | 3.5 | 0.95 | 0.44 | NS |
| Wi-Fi facilities | 3.49 | 3.59 | 3.51 | 3.44 | 4 | 0.59 | 0.67 | NS |
| Tourist transport | 3.46 | 3.6 | 3.77 | 3.88 | 4 | 1.21 | 0.31 | NS |
| Environmental literacy | 3.63 | 3.38 | 3.55 | 3.56 | 3.83 | 1.5 | 0.2 | NS |
| Nature conservation | 3.49 | 3.42 | 3.64 | 3.5 | 3.5 | 0.68 | 0.6 | NS |
| Monument preservation | 3.43 | 3.63 | 3.62 | 3.25 | 3.67 | 1.1 | 0.35 | NS |
| Community clean | 3.63 | 3.63 | 3.55 | 3.38 | 4 | 0.44 | 0.78 | NS |
| Littering | 3.73 | 3.69 | 3.66 | 3.44 | 3.83 | 0.22 | 0.93 | NS |
| Foreign threat | 3.75 | 3.68 | 3.51 | 3.5 | 3.33 | 0.75 | 0.56 | NS |
| Noise garbage | 3.65 | 3.56 | 3.43 | 3.5 | 4.33 | 1.28 | 0.28 | NS |
| Fume exhaust | 3.67 | 3.77 | 3.68 | 3.69 | 3.67 | 0.27 | 0.9 | NS |
| Water source | 3.62 | 3.72 | 3.72 | 3.5 | 3.5 | 0.41 | 0.8 | NS |
| Habitat | 3.68 | 3.62 | 3.38 | 3.31 | 3.5 | 1.1 | 0.35 | NS |
| Vegetation | 3.35 | 3.6 | 3.4 | 3.44 | 3.17 | 1.57 | 0.18 | NS |
| Air quality | 3.71 | 3.73 | 3.72 | 3 | 3.67 | 0.08 | 0.99 | NS |
| Lake water quality | 3.43 | 3.55 | 3.3 | 3 | 3.67 | 1.98 | 0.1 | NS |

The development and current environmental status of each region in Taiwan have their own characteristics, but the environmental quality requirements of people in different places of residence are different. However, due to people's awareness of maintaining a high quality of life and environment and the implementation of local governments' high standard of policy decisions related to environmental maintenance, there was a slight difference in the perceptions of people living in different places about the environmental development of the tourist area. However, the difference was not significant ($p > 0.01$).

## 4. Results and Discussion

Few studies on tourism behaviors have been conducted from the perspectives of regional differences and the concept of space. The answer can be obtained if the research is carried out based on travel experiences [28,29] and the related analysis is conducted from the perspective of space [30,31]. Therefore, a comprehensive decision on the sustainable development of the reservoir water source area on the island can be obtained if the questionnaire results are first examined using the statistical validation method, then the interview method is used to analyze the implications presented, and finally, the multivariate verification method is used for summarization and exploration.

### 4.1. Northern Tourists' Perception of the Development Status of the Water Source Area of the Outlying Island Reservoir

In the northern region, the technology industry has been well developed. There are high-rise buildings everywhere, the pace of life is fast, but there is a little relaxed atmo-

sphere of tourism and beautiful natural scenery. In the water source area, the air is clean, the lake scenery is beautiful, and the characteristics of buildings and commercial goods, consumption and life experiences are different from those in cities. Therefore, we believe that the air quality in the region is good, the interaction and community participation are strong, and the cultural activities, entrepreneurial opportunities, functional mix, and cultural and creative products are abundant.

However, the northern region has a wide variety of consumer goods, adequate medical facilities and public health awareness, modern construction facilities that are ahead of the rest of Taiwan, and perfect transportation facilities. Because there is only one external road in the scenic area, the number of tourists has decreased greatly, the frequency of public transportation has been reduced, and medical resources are lacking. Therefore, they believe that the current tourism transportation, preservation of monuments and trails, tourism organizations, and garbage cans are not enough. They believe that the work and leisure opportunities, healthcare, and quality of life are not adequate, the environmental quality of tourists is not good, the vegetation and lakes are damaged, and the ecology is threatened. As a result, the desire for return visits is low.

### 4.2. Central Tourists' Perception of the Development Status of the Water Source Area of the Outlying Island Reservoir

Due to the short distance of the journey, the tranquil environment of the water source area, good air and water sources, product characteristics, perfect parking and rest facilities, trails, signs and public security planning, not only can different travel experiences be obtained, but also there are business opportunities for entrepreneurs to grasp. Therefore, at present, the local air quality is good, public security management, indicators, parking and leisure, trails, noise and waste management measures are sufficient, vocational training and entrepreneurial opportunities are high, and functional combinations and cultural and creative products are abundant.

However, tourists in the central region can visit more frequently due to the proximity to scenic spots and the advancement of technology, perfect leisure sports and transportation facilities, and active commercial trade. Moreover, people's awareness of leisure and environmental quality requirements, high-quality life requirements, small differences in local products, and high industry similarities have caused the region's attractiveness to the people in the central region to gradually disappear. Therefore, they believe that the area's popularity is declining, tourism facilities and architectural style are not distinctive enough. They believe that wages are low, transportation connections, community communication and public interaction are lacking, environmental quality, littering, grease, and smoke emissions are problematic. The quality of lake water resources is affected, and the living space is poor.

### 4.3. Southern Tourists' Perception of the Development Status of the Water Source Area of the Outlying Island Reservoir

In the southern region, the industries have been well developed, industries and tourism resources have been diversified, but the competition for jobs has been intense. Due to the characteristics of local industries and commercial goods, the complete living space and transfer planning, the good interaction between the community and enterprises, the high popularity, and the plenty of business opportunities for entrepreneurs to grasp are the key factors. Therefore, they believe that the local area currently has good air and water quality, distinctive architectural styles, diversified tourism transportation, cultural activities, tourism industry and cultural and creative products, active public interaction, comfortable tourism space, many vocational training opportunities, great entrepreneurial opportunities, and high visibility.

Nevertheless, because the south is known as the mecca of democracy in Taiwan, communication with the public is good, and public opinion is valued. Moreover, industrial technology has advanced, tourism resources and leisure facilities are perfect, urban sports and consumer choices are diverse, and leisure and health and welfare measures are ad-

equate. However, local tourist attractions, facilities and activities have existed for many years, with small differences and little changes. Therefore, they believe that the region's current protection policies and corporate image are insufficient, community communication, tourism facilities, parking and resting, and bicycle rental measures are not satisfactory. They believe that leisure options and wages are low, and people's environmental literacy is poor, with littering and insufficient community cleanliness affecting the vegetation and lake environment, resulting in a lower desire to revisit.

Although the economic level and development effectiveness of central Taiwan are on par with those of the south, the south is rich in history and cultural architecture and compared to the central region, people in the south are farther away from the reservoir water sources, visit less frequently, and have lower travel imagery. Therefore, people in the south are more concerned about the architectural style and cultural characteristics of scenic spots, the quality of tourism activities and services, and the popularity of scenic spots than people in the central region.

### 4.4. Eastern Tourists' Perception of the Development Status of the Water Source Area of the Outlying Island Reservoir

The development of the eastern region has been slow, the cultures of the two places are similar, there are few job opportunities, and the pace of life is slow due to the diversified organizations of local enterprises, living environment, industries, facilities, medical care, low consumption disputes, and complete leisure facilities and transfer planning. Therefore, they believe that the local tourism transportation and healthcare facilities are well developed, the tourism industry, interpretation facilities, and bicycle rental measures are well established. Public interaction, community participation, and cultural preservation are effective, expenditure costs are low, job opportunities are abundant, air and lake quality can still be maintained at a certain level, people's environmental literacy is satisfactory, damage by tourists is rare, and living space is more comfortable.

However, because foreign investors in the eastern region are keen on the real estate industry, the local population is small, and there are not enough tourism facilities, but the enterprises are willing to communicate with the people and the residents interact with each other enthusiastically and can complement each other in terms of resources or manpower shortage. Compared to Sun Moon Lake, there is not much change in the atmosphere, environment, and tourism resources in the east. Therefore, they believe that the awareness of local enterprises to give back and communicate with the community is low, tourist discounts are scarce, the quality of activities, services, and maintenance of monuments and trails are poor, and the emission of fumes affects the existing ecological habitat, and the willingness to revisit is low.

### 4.5. Outlying Island Tourists' Perception of the Development Status of the Water Source Area of the Outlying Island Reservoir

On offshore islands, there are relatively scarce land and industrial resources, and most people depend on tourism for much of their income, with a high level of competitiveness. Due to the tranquil environment, quality air and water, cultural and industrial characteristics, preferential measures, good tourism and living environment, complete leisure and Wi-Fi facilities, tourism guidance, public security, health and medical planning, diversified local culture and activities, high popularity, and the plenty of business opportunities for entrepreneurs to grasp. Therefore, they believe that the local air and lake quality is good, the traffic connection and bicycle path planning, tourism indicators, healthcare, and other measures are perfect. They believe that the tourism industry is diversified, there are many opportunities to start a business and work, the maintenance of culture and trash can installation is good, there are many tourism incentives, the community exchange is intensive, the public security management is perfect, and the visibility is high.

However, although there is little room for development in the outlying island, the number of tourists is high, and there is still room for growth in the tourism industry, and the local area retains buildings with local characteristics, foreign-invested enterprises have

integrated into the local culture, along with abundant marine tourism resources, the separation of tourist areas and residential areas, large diversity in the types of industries, strong interaction among residents, compensating for the manpower for industrial development, and little difference in the existing quality of life. Therefore, they believe that although the current industrial development in the local area is enthusiastic, there is little diversity in the type of industry. They believe that the income is low and the competition is high, coupled with the low willingness of enterprises to give back to the community, insufficient tourism policies and facilities, insufficient space, manpower and funds available. They believe that the planning of bicycle paths and tourism facilities are not adequate, changing the appearance of the existing ecology and vegetation, affecting the existing environment, tourism activities and quality of life, lowering consumer willingness, and decreasing the willingness to revisit.

The outlying islands have few economic resources, and tourism is one of the main sources of economy. With a small land area, fast travel speed, and high population interaction, the people in the outlying islands are more concerned about community exchange and are more enthusiastic about community and industry interaction than those in the eastern region. Therefore, the outlying islands are more sensitive to the current state of community exchange than the eastern regions.

## 5. Conclusions and Recommendations

The study found that, in the reservoir source area of the outlying islands, the quality of the natural environment, lakes, and water sources is gradually declining, medical facilities in the area are inadequate, transportation measures are poor, the types of industries, consumer goods, and attractions overlap, distinctive culture and architecture are disappearing. We also found that the amount of tourism waste is increasing, waste collection points are insufficient, and interaction between foreign companies and communities needs to be improved, which has affected tourists' desire to visit the area. Among them, the interactivity of the people, the smoothness of tourism routes, the quality of tourism activities and life, and the potential for entrepreneurship and business opportunities in the reservoir's water source area of the outlying islands are the motivations for tourism for people living in areas with fast-paced economic conditions and better quality of life. The level of consumption and the effectiveness of environmental preservation are the attractions for tourism for people who live in slow-paced regions and are looking to improve their economic conditions and quality of life. Good air and environmental quality are key to tourism, and the unique cultural, architectural and industrial appeal, as well as the search for job opportunities, are the main reasons for spending for people from all regions.

Increasing the variety of consumer goods, improving medical and transportation facilities, preserving unique culture and architecture, linking foreign-invested enterprises with community interactivity, enhancing local people's interaction, and compensating for the deficiencies in human resources for industrial development are the key points for future improvement of the reservoir water source area of the outlying islands. Based on the conclusions of the study, the following recommendations are proposed:

1.  Suggestions for environmental protection To improve the maintenance quality of the natural environment, lakes, and water sources, to improve the environmental quality of tourists and the quality of life in tourist areas, to reduce the amount of tourism waste, to plan convenient transportation connections, to design an APP system for regional public transport schedule query or taxi connection, and to build consensus among residents on tourism development and environmental maintenance should be the future policy indicators for the reservoir water source area of the outlying islands;

2.  Suggestions for management measures To increase the number of medical facilities and trash cans in scenic spots, to strengthen tourism indicators and environmental education slogans, to enhance corporate social responsibility, to distinguish different types and appearances of tourism activities or products, to design an online

regional food guide APP, and to pay attention to public opinion should be the future improvement policies of the authorities;

3.　Suggestions for industry development To categorize industries, diversify enterprises, use culture and architecture, and develop leisure and sports activities or facilities should be the future development goals of local industries.

4.　Suggestions for future research To extend the research methodology and explore the current development situation of the area, which expects diversified development despite the lack of water source. Analyze the differences in the perceptions of different backgrounds, ages, and occupations of the reservoir water areas in the outlying islands. To extend the study to explore the attractiveness of tourism behaviors of people with different residential backgrounds and analyze tourism path patterns.

**Author Contributions:** Conceptualization, J.-H.Y. and H.-H.L.; methodology, H.-H.L. and C.-H.H.; software, C.-H.H.; validation, Y.-C.L. and Y.-C.L.; analysis, H.-H.L.; investigation, K.-C.T.; resources, Y.-C.L.; data curation, K.-C.T.; writing—original draft preparation, H.-H.L. and C.-H.H.; writing—review and editing, H.-H.L.; visualization, H.-H.L.; supervision, H.-H.L.; project administration, C.-H.H.; funding acquisition, C.-H.H. and J.-H.Y. All authors have read and agreed to the published version of the manuscript.

**Funding:** This research received no external funding.

**Institutional Review Board Statement:** All subjects in the study were anonymously labeled and agreed to participate in the survey.

**Informed Consent Statement:** Informed consent was obtained from all subjects involved in the study.

**Data Availability Statement:** No data support.

**Conflicts of Interest:** The authors declare no conflict of interest.

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
