# Peer review of "Research on Sustainable Development Evaluation of Reservoir Water Source Area in Island Area"

_water, doi:10.3390/w13081130_

Round 1

Reviewer 1 Report

This manuscript by author provides an interesting subject about Research on sustainable development evaluation of reservoir water source area in island area

, and I can say the same things about the manuscript.

Detailed comments:

  1. What is innovation of this M.S., author should give the more innovation clearly in introduction part? And there are many related researches, what is the news?

  1. The results are very good, but not just the data relationship. It is suggested to add some mechanism analysis, for example, the poor interaction between businesses and communities has led to an overlap in the types of industries; consumer goods; and attractions; and a lack of transportation and medical facilities; which affects people's desire to travel, the author can supplement the analysis of mechanism.

Author Response

Review 1

  1. What is innovation of this M.S., author should give the more innovation clearly in introduction part? And there are many related researches, what is the news?

 Thank you reviewers for their suggestions

We have added instructions, such as line 128-134.

  1. The results are very good, but not just the data relationship. It is suggested to add some mechanism analysis, for example, the poor interaction between businesses and communities has led to an overlap in the types of industries; consumer goods; and attractions; and a lack of transportation and medical facilities; which affects people's desire to travel, the author can supplement the analysis of mechanism.

Thank you reviewers for their suggestions

We have made supplements in 4. Results and Discussio and discussed appropriately in 5. Conclusions and Recommendations.

Reviewer 2 Report

I think the article entitled "Research on sustainable development evaluation of reservoir water source area in island area" is very interesting and I suggest to the editor that it be published. I have only one suggestion, which in my opinion would greatly improve the article in terms of originality and scientific soundness: the authors could improve the article by framing the paper in the perspective of smart and sustainable islands, trying to analyze how, from a theoretical point of view, development in closed contexts such as islands can be triggered. 

In this regard I suggest you read and quote the following articles:

  1. Di Staso, U., Magliocchetti, D., & De Amicis, R. (2014). Smart-Islands: Enhancing user experience for mediterranean islands for tourism support. In International Conference of Design, User Experience, and Usability (pp. 223-233). Springer, Cham.
  2. Garau, C., et al. (2020). Territorial cohesion in insular contexts: assessing external attractiveness and internal strength of major Mediterranean islands. European Planning Studies, 1-20.
  3. Korak, B., & Shivendra, P. (2020). Sustainable smart specialisation of small-island tourism countries. Journal of Tourism Futures6(2), 121-133.
  4. Xu, C., Li, X., & Wu, X. (2020). Evaluation of Island Tourism Sustainable Development in the Context of Smart Tourism. Journal of Coastal Research103(SI), 1098-1101.

Author Response

Review 2

I think the article entitled "Research on sustainable development evaluation of reservoir water source area in island area" is very interesting and I suggest to the editor that it be published. I have only one suggestion, which in my opinion would greatly improve the article in terms of originality and scientific soundness: the authors could improve the article by framing the paper in the perspective of smart and sustainable islands, trying to analyze how, from a theoretical point of view, development in closed contexts such as islands can be triggered. 

Thank you reviewer

We have added relevant content. Such as the discussion content of line 128-142 and 5. Conclusions and Recommendations.
